# Experimental confirmation of driving pressure boosting and smoothing for hybrid-drive inertial fusion at the 100-kJ laser facility

Ji Yan[1,6], Jiwei Li[2,3,6], X. T. He [2,3] ✉, Lifeng Wang[2,3], Yaohua Chen[2], Feng Wang[1], Xiaoying Han[2], Kaiqiang Pan[1], Juxi Liang[1], Yulong Li[1], Zanyang Guan[1], Xiangming Liu[1], Xingsen Che[1], Zhongjing Chen[1], Xing Zhang[1], Yan Xu[2], Bin Li[2], Minqing He[2], Hongbo Cai [2,3], Liang Hao[2], Zhanjun Liu [2,3], Chunyang Zheng[2,3], Zhensheng Dai[2], Zhengfeng Fan [2], Bin Qiao [3,4], Fuquan Li[1], Shaoen Jiang[1], M. Y. Yu[5] & Shaoping Zhu [1,2]

In laser-driven inertial confinement fusion, driving pressure boosting and smoothing are major challenges. A proposed hybrid-drive (HD) scheme can offer such ideal HD pressure performing stable implosion and nonstagnation ignition. Here we report that in the hemispherical and planar ablator targets installed in the semicylindrical hohlraum scaled down from the spherical hohlraum of the designed ignition target, under indirect-drive (ID) laser energies of ~43–50 kJ, the peak radiation temperature of $200 \pm 6$ eV is achieved. And using only direct-drive (DD) laser energies of 3.6–4.0 kJ at an intensity of $1.8 \times 10^{15}$ W/cm$^2$, in the hemispherical and planar targets the boosted HD pressures reach 3.8–4.0 and 3.5–3.6 times the radiation ablation pressure respectively. In all the above experiments, significant HD pressure smoothing and the important phenomenon of how a symmetric strong HD shock suppresses the asymmetric ID shock pre-compressed fuel are demonstrated. The backscattering and hot-electron energy fractions both of which are about one-third of that in the DD scheme are also measured.

Laser-driven inertial confinement fusion (ICF) is an important way to obtain fusion energy by converting laser energy to a driving pressure imploding compression fuel performing ignition and burning under the support of the fuel motion inertia. So, finding a sufficiently high driving pressure with good smoothing for ICF is a top priority as well as a great challenge. Once such pressure is obtained, we can artistically design the ignition target to perform the stable implosion and achieve ignition.

The indirect-drive (ID)[1] and the direct-drive (DD)[2] schemes by high-temperature ablation pressure driving implosion have been applied to explore the deuterium-tritium (DT) fuel ignition and

burning for decades. In these schemes, a typical spherical capsule is made up of an ablator (the outer shell), the DT ice, and a central low-density DT gas region. The driving pressure at the ablator surface is generated by directly absorbed laser energy or by indirectly absorbed thermal X-rays, which are converted by ID laser energy incident on a high-$Z$ inner wall of the hohlraum containing the capsule inside. The occurrence of the implosion for both ID and DD can be described by the rocket model, where the rapid outward expansion of the high-temperature ablated ablator results in low coronal plasma density, and hence the low ablation pressure drives the fuel implosion inward and the hotspot ignition in the stagnation time.

[1]Laser Fusion Research Center, China Academy of Engineering Physics, 621900 Mianyang, P. R. China. [2]Institute of Applied Physics and Computational Mathematics, 100094 Beijing, P. R. China. [3]Center for Applied Physics and Technology, Peking University, 100871 Beijing, P. R. China. [4]School of Physics, Peking University, 100871 Beijing, P. R. China. [5]College of Engineering Physics, Shenzhen Technology University, 518118 Shenzhen, P. R. China. [6]These authors contributed equally: Ji Yan, Jiwei Li. ✉e-mail: xthe@iapcm.ac.cn

The ID scheme has been used for experiments on the National Ignition Facility (NIF)[3,4], in which the target consists of a cylindrical hohlraum and a layered fuel capsule inside. The thermal X-rays (radiation) with radiation temperature $T_r$ ~ 300 eV ablate the surface of the ablator generating the radiation ablation pressure ~100 Mbar[4] for the plastic (CH) ablator and ~134 Mbar[5] for the high density carbon (HDC) ablator. These radiation ablation pressures have non-negligible nonuniformities[6,7], and drive the ID shocks asymmetrically imploding fuel with the maximal implosion velocity of less than 400 km/s, resulting in severe hydrodynamic instabilities to affect the stable ignition. It was recently announced that with 2.05 MJ laser energy the great achievement of the fusion energy gain of ~1.5 was realized experimentally, but unfortunately due to unstable implosion, the result can not be repeated easily.

For the DD scheme[2,8], laser energy is directly absorbed near the critical surface without using the hohlraum. The fuel is imploded by shocks driven by the electron ablation pressure converted by DD laser energy, and compressed to a high density with the central hotspot for ignition, like in ID. Its advantage is the high efficiency of laser energy to the capsule, but its shortcomings are the implosion asymmetry, caused by the overlapping of laser beams near the critical surface, resulting in hydrodynamic instabilities. If one would like to suppress hydro-dynamic instabilities by boosting the electron ablation pressure it may lead to severe laser-plasma instability[9,10] due to the electron ablation pressure being proportional to $I_L^{2/3}$, where $I_L$ is the DD laser intensity.

The hybrid-drive (HD) scheme, a coupling of ID and DD, in which the target consists of a spherical hohlraum rather than the cylindrical hohlraum and a layered DT fuel capsule with a CH ablator inside the spherical hohlraum, was proposed[11,12] to provide an ideal HD pressure realizing the stable implosion and performing the hotspot ignition at the nonstagnation (before stagnation) time with the low convergence ratios and the suppression of hydrodynamic instabilities. In the whole HD implosion process, the ID laser with the pre-pulse and main pulse continuously enters the spherical hohlraum through laser entrance holes (LEHs)[13,14] and is absorbed on the inner wall converting into thermal X-rays as schematically plotted in Fig. 1a. In the first phase, only the ID pre-pulse laser works. Due to the larger mass ablation rate ($\dot{m} \propto T_r^3$ for the CH ablator[15]) by radiation, the ablator surface produces a long-scale-length ID corona plasma while the fuel in the capsule is pre-compressed by the asymmetric ID shocks driven by the nonuniform radiation ablation pressure. In the second phase, the ID main-pulse laser continuously offers the long-scale-length corona plasma and further enhances the pre-compression of the fuel. Meanwhile, the DD laser with intensity of $I_L$ ~ 1 − 2 PW/cm² (1 PW = 10¹⁵ W) entering the spherical hohlraum along the opposite direction of the radius of the capsule is absorbed near the ID laser pre-offered critical

surface and is converted to a supersonic-electronic-heat wave. We find that in the ID corona plasma, as long as the large enough distance $\Delta R_{ID}$ between the radiation ablation front and the critical surface is greater than close to a slowing-down length $d_s = \int_0^{\Delta t} v_e dt$, this supersonic-electronic-heat wave propagating in $\Delta R_{ID}$ can slow down to a sonic speed before reaching the radiation ablation front and a plasma compressive wave following a precursor shock is formed, where $v_e$ is a supersonic-electronic-heat wave velocity and $\Delta t$ is the slowing down duration. This compressive wave with a high plasma pressure produces an effect under the stable support of the DD laser, similar to a "bulldozer", to thermally compress the low ID corona plasma density $\rho_a$ near the radiation ablation front into sufficiently high HD plasma density $\rho_{HD}$ ($\gg \rho_a$) to form an HD density plateau between the compression wave front and the radiation ablation front, where the HD pressure $P_{HD} = \Gamma \rho_{HD} T_r$ by increasing the plasma density is boosted much higher than the radiation ablation pressure $P_a = \Gamma \rho_a T_r$ with $\Gamma$ the ideal gas pressure constant. We find from numerical simulations that if $d_s$ matches with $\Delta R_{ID}$ well, in the density plateau there are the fitted hydroscaling relations of the maximal HD pressure $P_{HD} \propto E_{DD}^{1/4} T_r$ and the maximal HD density $\rho_{HD} \propto E_{DD}^{1/4}$, where $E_{DD}$ is the DD laser energy. On the other hand, during the supersonic-electronic-heat wave slowing down, its pressure nonuniformity $\delta P/P$, with the perturbation wave length $\lambda_p = 2\pi R_{cr}/\ell$ caused by overlapping of DD laser beams near the critical surface of radius $R_{cr}$, is continuously weakened and thermally smoothed very well in the form $\delta P/P \approx 2/3(\delta I_L/I_L)Exp(-2\pi\beta\Delta R_{ID}/\lambda_p)$ when $2\pi\beta\Delta R_{ID} > \lambda_p$, where $\ell$ is the perturbation mode number and $\beta = 1.5 − 2$ is a transverse thermal-ablation correction factor from 2D simulation[11,12,16].

So, the bulldozer and thermal smoothing effects change the radiation ablation pressure into a smoothed HD pressure much greater than the radiation ablation pressure. A strong symmetric HD shock driven by the ideal HD pressure rapidly entering the imploding capsule collides in the opposite directions with the asymmetric relatively weak ID shock which is reflected from the center of the hotspot after pre-compressed the fuel and just arriving at the interface of the hotspot, and then the ID shock reflected inward becomes weaker and is quickly caught up, swallowed and merged by the strong HD shock. Thus, the asymmetric ID shock in the early implosion stage is suppressed to prevent further asymmetric implosion. Then, instead of the ID shock, the HD shock further symmetrically compresses the fuel and suppresses hydrodynamic instability to perform the hotspot ignition at the nonstagnation time with the convergence ratio $C_r < 25$.

We have to explain why in the DD scheme there is no "bulldozer" effect because, in the HD scheme, before the DD laser arrives the thermal X-rays with the large mass ablation rate provided a long enough distance $\Delta R_{ID}$ between the radiation ablation front and the critical

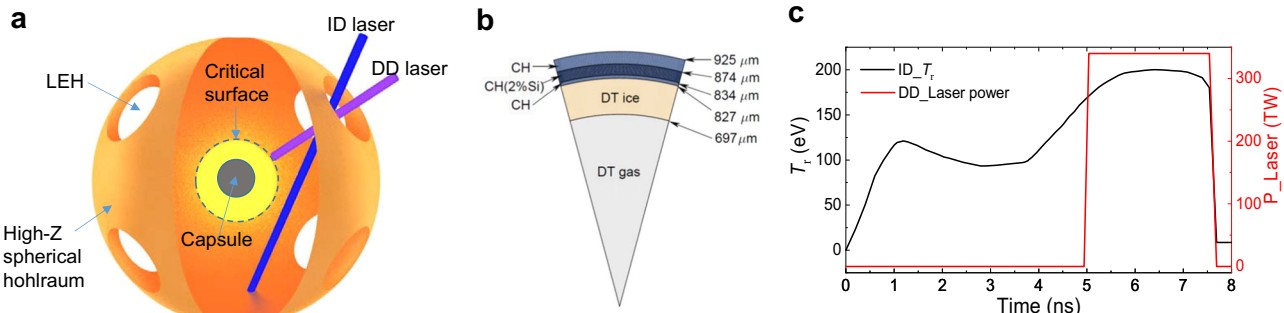

**Fig. 1 | Schematic ignition target, the DD laser power (red) and radiation temperature $T_r$ converted by the ID laser (black). a** The spherical hohlraum with an inner radius of 5 mm has 8 LEHs of each radius of about 1.6 mm and the ID laser (blue) and DD laser (purple) enter the hohlraum from LEHs, where the incident lasers are plotted schematically only in one of LEHs. The capsule is put in the center of the hohlraum. The critical surface is plotted by the dashed line and $\Delta R_{ID}$ between the critical surface and radiation ablation front (radiation ablation front) is the yellow region. **b** The layered DT fuel capsule structure and sizes. **c** Radiation temperature $T_r$ (black) is converted by ID laser on the high-$Z$ inner wall of the spherical hohlraum, and DD laser power (red) vs time.

surface. While, in the DD scheme, there is no large enough slowing down distance prepared in advance, and the distance between the electron ablation front and the critical surface is too short due to the low mass ablation rate by electrons so that the supersonic-electronic-heat wave directly hits on the capsule with no time to slow down.

With one- and two- dimensional radiation hydrodynamic codes-LARED series[17] to a large number of numerical simulations[11,12,16,18] for kinds of modeling HD ignition targets consisting of a layered capsule with the CH ablator and a spherical hohlraum with 5-mm radius and 8 LEHs ($P_4$ asymmetry at mode number $\ell = 4$) in Fig. 1a, we have studied HD pressure boosting, smoothing, asymmetric ID shock suppressing, and the symmetric HD shock performing implosion compression and hotspot ignition, and the results are partly listed in Table of the "Methods" section (see below).

We present some simulation results of the layered capsule with a DT mass of 0.245 mg and a CH ablator with an outer radius of 0.925 mm (Fig. 1b).

Using ID laser energy $E_{ID}$ = 900 kJ with a pre-pulse of about 5 ns and a main pulse of about 3 ns and DD laser energy $E_{DD}$= 825 kJ with a flattop pulse of 2.5 ns launching in the ID main pulse stage, as shown in Fig. 1c, simulations by one-dimension LARED code show that the ID laser energy offers peak radiation temperature $T_r$ = 200 eV (corresponding radiation ablation pressure $P_a \approx 45$ Mbar) and $\Delta R_{ID} \sim 300$ μm, and the DD laser intensity about 1.8 PW/cm$^2$ offers the slowing down length $d_s \sim 270$ μm $< \Delta R_{ID}$ with an averaged slowing down velocity of $v_e \sim 800$ km/s. When the rarefaction wave caused by DD laser unloading reaches the front of the HD density plateau, which has a width of about 50 μm, the "bulldozer" effect has compressed the low ID corona density $\rho_a$ into a high HD density $\rho_{HD} \sim 4.85$ g/cm$^3$ and boosted the maximal HD pressure to $P_{HD} \sim 820$ Mbar, about ~19 times the radiation ablation pressure. It must be explained that in this HD density plateau, radiation temperature $T_r$ elevates from 200 eV to 270 eV caused by inverse bremsstrahlung absorption because of the increased optical thickness when HD density is over 3 g/cm$^3$, and therefore, the HD pressure $P_{HD}$ increases as well. On the other hand, the pressure non-uniformity of the supersonic-electronic-heat wave propagating from the critical surface toward the radiation ablation front is thermally smoothed. Taking the laser intensity nonuniformity $\delta I_L/I_L \sim 2\%$ on the critical surface of the radius $R_{cr} \sim 800$–$900$ μm, $\Delta R_{ID} \sim 300$ μm, $\beta = 1.5$, $\ell = 4$ ($P_4$), we have $\delta P/P \sim 0.17\%$ at the radiation ablation front. The simulation by 2D code with 3D ray tracing shows that using a continuous phase plate (CPP) with a diameter of 1.5 mm, $\delta P/P \sim 0.2\%$ in excellent smoothing is obtained.

A stronger symmetric HD shock, which is driven by the boosted and smoothed HD pressure $P_{HD}$, provides a maximal implosion velocity over 426 km/s by the bulldozer effect rather than by the rocket effect like in ID and DD, in the early implosion stage of the convergence ratio of about 5, suppressed the asymmetric ID shock and hydrodynamic instabilities caused by the ID shock. Then, the symmetric HD shock steadily implodes the capsule and drives the hotspot non-stagnation ignition in the convergence ratio $C_r \approx 24$, releasing fusion energy of about 20 MJ by total laser energy of 1.725 MJ with the fusion energy gain over 11. The 2D fusion energy gain is close to the one-dimensional result.

In addition, we also see in the Table of the Methods section (see below) that for the capsule with a radius of 916 μm containing DT fuel of 0.231 mg, even if the total laser energy is 1.49 MJ the system can still provide the maximal HD pressure as high as 770 Mbar and the maximal implosion velocity of 415 km/s to achieve the fusion energy gain close to 10 under the convergence ratio of 24.

An early preliminary experiment[18] was performed at the SG-III facility[19] with laser energy of ~100 kJ at a wavelength of 0.35 μm. A planar disc-like sandwich target, consisting of CH-Al-SiO$_2$ in which CH is the ablator, is installed in the bottom of a semicylindrical hohlraum with the top opening for LEH. The ID laser energy of ~43 kJ incident on

the inner wall of this hohlraum offers a peak radiation temperature of 200 eV, which rapidly heats the surface of the CH ablator producing a large enough distance, $\Delta R_{ID}$, between the radiation ablation front and the critical surface. Then, under such ID corona plasma, the DD laser energy of 3.6 kJ is absorbed near the critical surface and converted to the bulldozer effect. The experimental result of the boosted HD pressure of 150 Mbar, about 3.0 times the radiation ablation pressure, is achieved, which successfully demonstrates HD pressure boosting in a planar ablator target.

Here, under the HD scheme, using the hemispherical and planar ablator targets installed in the semicylindrical hohlraum that is scaled down from the spherical hohlraum in the designed ignition target to perform the experiments on SG-III, we report the experimental results of HD pressure boosting, smoothing, ID shock suppressing, and laser-plasma interaction (LPI).

## Results

### Experimental target design

Due to the limitation of laser energy on SG-III, the spherical hohlraum could not provide a high enough radiation temperature to meet the necessary conditions for the "bulldozer" effect, and therefore, the experimental target is designed in two steps. In the first step, using the ID laser energy balance relationship $\eta E_{ID} \approx S\sigma T_r^4 (1 - \varepsilon)\tau$, the spherical hohlraum with a 5-mm radius and 8 LEHs (Fig. 1a) in the ignition target is scaled down to a semicylindrical hohlraum with single LEH under keeping radiation temperature $T_r$ = 200 eV and $d_s < \Delta R_{ID} \sim 300$ μm unchanged, where $\eta$ is the conversion efficiency for thermal X-rays, $S$ is the total hohlraum area, $\varepsilon$ is the area ratio of LEHs to the hohlraum, $\sigma$ is the Stefan–Boltzmann constant, $\tau$ is the effective ID laser pulse duration, and radiation energy absorbed at the ablator surface less than (4–10)% has been ignored. Taking $\varepsilon \sim 20\%$ in the semicylindrical hohlraum the same as in the spherical hohlraum, the inner wall area of the semicylindrical hohlraum can be written in the form $(S)_{ex} = (S)_{ig} \times (E_{ID}/\tau)_{ex} / (E_{ID}/\tau)_{ig}$, where the subscripts "ex" and "ig" denote the quantities for the experiment and ignition respectively. Using ID laser energy $(E_{ID})_{ig}$=900 kJ with the effective $\tau_{ig} \approx 3.5$ ns (a pre-pulse and a main pulse, as seen in Fig. 1c) and the spherical hohlraum area $(S)_{ig}$=100 $\pi$ mm$^2$ (the radius of 5 mm) in the ignition target, and ID laser energy $(E_{ID})_{ex} \sim 50$ kJ with main pulse duration $\tau_{ex} \approx 3$ ns in Fig. 2b for the experiment, we acquired the semicylindrical hohlraum approximately at a diameter of 2.5 mm and a length of 1.5 mm (Fig. 2c), which has the equivalent spherical hohlraum diameter of about 2.62 mm. So, this semicylindrical hohlraum can naturally provide the same radiation temperature $T_r$ = 200 eV as the spherical hohlraum in the ignition target to rapidly heat the CH ablator surface.

In the second step, under the radiation temperature of $T_r$ = 200 eV and the DD laser energy of 3.6 kJ and 4.0 kJ, as plotted in Fig. 2b by red color, we use a hemispherical target with the spherical convergence implosion and a planar target without the spherical convergence implosion, in which both targets are installed in the same size semicylindrical hohlraum, as a comparison to investigate the HD pressure boosting and smoothing effects, these are at the heart of the HD scheme. The hemispherical target, which is mounted in a hole with a radius of 475 μm at the bottom end of the semicylindrical hohlraum of the top end opening for LEH, as seen in Fig. 2c, contains a hemispherical CH ablator shell with a radius of $R_a$= 620 μm and a thickness of 80 μm, followed by a 40 μm aluminum (Al) shield for whittling the M-band emission from the gold hohlraum wall and the energetic electrons from the laser-plasma interaction, with a 500 μm radius quartz window for VISAR light diagnosis, and the spherical crown surface of the CH ablator with an open angle of 100° is irradiated by thermal X-rays, as shown in Fig. 2c. The planar target with a disk-like layered structure of a diameter of 800 μm contains a 70 μm CH layer serving as the ablator, followed by a shield of a 30 μm molybdenum (Mo) for the DD laser energy of 4.0 kJ or a 40 μm Al for the laser energy

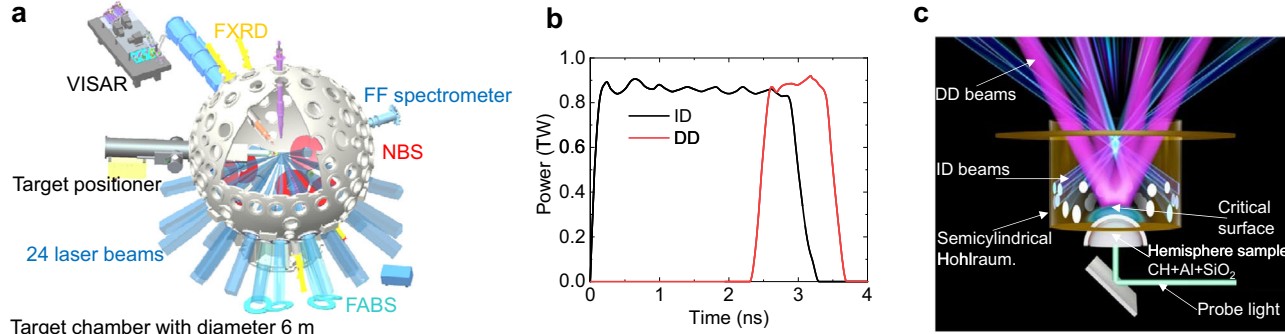

**Fig. 2 | Schematic of the HD experiments. a** The target chamber has a diameter of 6 m, Velocity Interferometer System with Any Reflector (VISAR) is for measuring the shock velocities, the flat-response x-ray diode (FXRD) is for measuring the radiation temperature $T_r$, the filtered fluorescence (FF) spectrometer is for measuring the hot-electron energy, the nearby backscatter station (NBS) and the full aperture backscatter station (FABS) are for measuring the backscattered laser energy. **b** Laser power for ID (black) and DD (red) vs time. **c** The hemispherical target, consisting of CH, Al, and quartz ($SiO_2$) layers, mounted in a hole in the bottom of the semicylindrical hohlraum. The ID lasers (blue) and the DD lasers (pink) enter the hohlraum through LEH, and the VISAR light arrives in the quartz layer via the reflector, which all are monitored by a static imaging system in the target chamber.

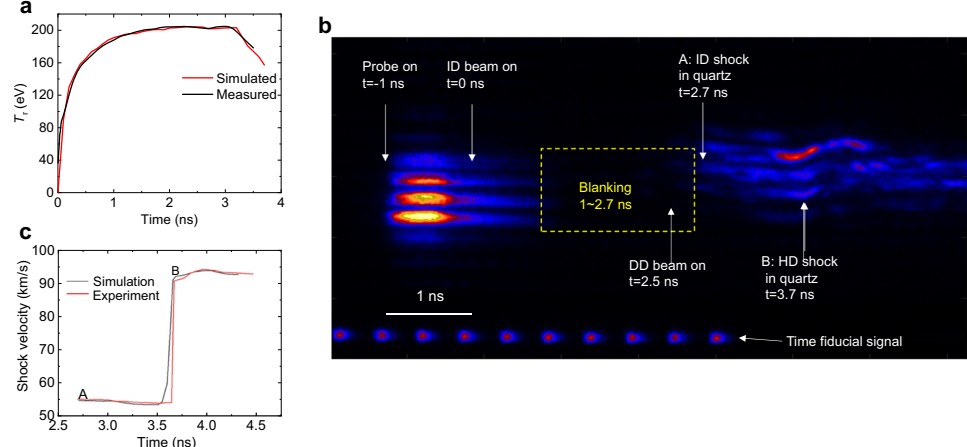

**Fig. 3 | Radiation temperature and shock velocities. a** The measured (black) and simulated (red) radiation temperature vs time. **b** Interferometer time-resolved image with VISAR, where the VISAR light tracks, "blanking" region, and perturbations A and B are shown. **c** Comparison between experiment (red) and simulation (black) for shock velocities from ID (A) and HD (B).

of 3.6 kJ, and then a 130 μm quartz window for VISAR light diagnosis, as shown in Fig. 2c if the hemispherical target in the semicylindrical hohlraum is replaced by the planar target and the hemispherical critical surface is replaced by the planar structure.

As previously mentioned, as long as the necessary conditions, involving radiation temperature $T_r = 200$ eV to ensure $d_s$ less than and close to $\Delta R_{ID} \sim 250-300$ μm, are met, a compressive wave can be produced to isothermally compress the ID corona plasma and to drive the density increase in the density plateau, leading to HD pressure boosting and smoothing.

We first discuss the hemispherical target experiment in detail. A spherical convergence effect on the density plateau further enhances the HD density and pressure. The shocks driven by the radiation ablation pressure and the HD pressure enter the layered hemispherical target in sequence and propagate convergently there.

In our experiments, VISAR[20] (explained in the Methods section) is used to record the evolution of the shock velocities in the quartz window, and the flat-response x-ray diode (FXRD)[21] is used to measure radiation temperature $T_r$ in the hohlraum. The setup on the target chamber is schematically plotted in Fig. 2a. In the experiment, beginning at time $t = 0$ the twenty 3-ns ID laser beams with an intensity of ~0.4 PW/cm² and total laser energy of $(52 \pm 3)$ kJ, through LEH in two rings at 49.5° and 55° with the hohlraum axis respectively (blue in schematic

Fig. 2c), are incident on the inner wall of the semicylindrical hohlraum filled a $C_5H_{12}$ gas with the density of 1.2 mg/cm³. The VISAR probe light turns on at $t = -1.0$ ns (Fig. 3b). During the whole implosion stage $0 < t < \sim 3.5$ ns, the ID lasers acting on the inner wall of the hohlraum continuously generate thermal X-rays with radiation temperature $T_r$ that ablates the CH ablator surface to pre-compress the hemispherical target. The heated CH surface rapidly expands upward in the axial direction of the hohlraum forming a low-density ID corona plasma with a long scale length. In the last ID stage, beginning at $t \sim 2.5$ ns, four DD laser beams of a 1.0-ns flattop duration (Fig. 2b) with the laser energy of $(4.0 \pm 0.2)$ kJ and intensity of ~1.8 PW/cm² in 28.5° (pink in Fig. 2c) entering the hohlraum from LEH through a CPP with the diameter of 500 μm propagate in the ID corona plasma provided in advance by ID laser and are absorbed near the critical surface (Fig. 2c), where the converted supersonic-electronic-heat wave propagating toward the radiation ablation front continuously slows down and smoothes.

## Radiation temperature

As we mentioned in the experimental target design, the inner-wall area of the semicylindrical hohlraum is far greater than that of the hemispherical or planar targets, therefore, for the ID laser energy $E_{ID} = 43 - 52$ kJ, the radiation temperature ($T_r \propto \sim E_{ID}^{1/4}$) in semicylindrical hohlraum for both targets should be the same in the range

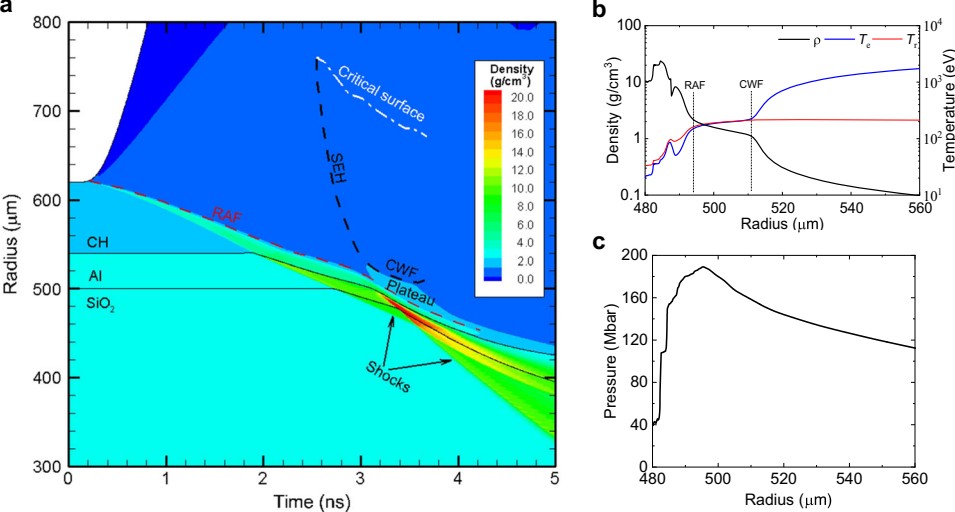

**Fig. 4 | The one-dimensional simulation results under the experimental parameters. a** Temporal trajectories of the supersonic-electronic-heat (SEH) wave, compression wave front (CWF), radiation ablation front (RAF), shock propagation in quartz, in which DD laser launching at $t = 2.5$ ns is absorbed near the critical surface. At $t \sim 3.5$ ns, close to the end of the DD laser, the space distributions: **b** electron temperature $T_e$ and radiation temperature $T_r$, and **c** the HD pressure and plasma density within the density plateau between compression wave front and radiation ablation front.

of the experimental errors. We have measured the history of ID radiation temperature with a peak of $T_r = (203 \pm 6)$ eV in the semi-cylindrical hohlraum by FXRD, and it agreed with the simulation result, as plotted in Fig. 3a.

### Shock velocities in the hemispherical target

The VISAR signals indicating the shock velocities are recorded in the interferometer time-resolution image (Fig. 3b). Until $t \sim 1.0$ ns, the probe light from VISAR is uniformly distributed in the quartz window. From $t \sim 1.0$ to $\sim 2.7$ ns, the quartz window exhibits weak "blanking", which can be attributed to M-band X-rays and energetic electrons penetrating through the CH and Al layers preheating the quartz window. The strong shock front produces a sharp density gradient in the quartz because the shield layer has prevented sufficient M-band emission and energetic electrons, and thus the VISAR light is significantly reflected so that we can deduce the shock velocities from the perturbations. At $t \sim 2.7$ ns, the probe light is first perturbed (point A in Fig. 3b) by the ID shock, and its velocity of $(55 \pm 1.0)$ km/s in the quartz can be obtained as shown by the red color in Fig. 3c. Then, at $t \sim 3.7$, as the HD shock caught up with the ID shock the probe light is again strongly perturbed (point B) with a sudden large jump with the HD velocity of $(92.7 \pm 2)$ km/s in the quartz, as plotted by red color in Fig. 3c, which is much larger than the ID shock velocity, showing significant HD pressure boosting. In addition, we also see from Fig. 3c that the strong HD shock caught up with the relatively weak ID shock, and swallowed and merged it, which showed that the symmetry HD shock can prevent the unstable implosion from the asymmetric ID shock. These behaviors are essential to the HD scheme.

In addition to the experiment of the DD laser energy of ~4 kJ, using the same target and the ID laser energy of ~48.5 kJ, we also performed another shot with the DD laser energy of 3.6 kJ with a flattop pulse of 1 ns. In this round of the experiment, we obtained the radiation temperature of $T_r = (201 \pm 6)$ eV, and in the quartz the HD shock velocity of $(82.7 \pm 2)$ km/s.

### Shock velocities in the planar target

In the first shot with ID laser energy of ~43 kJ and DD laser energy of 4 kJ, the measured radiation temperature is $T_r = (200 \pm 6)$ eV. We obtained the ID shock velocity $(50 \pm 0.9)$ km/s in the quartz window. Then, as the HD shock caught up with the ID shock a sudden large jump in the HD

velocity appears, similarly to point B in Fig. 3c. Finally, due to the quartz layer's finite thickness of 130 μm, the HD shock breaks out on a 5° downward slope rear of the quartz window with a breakout velocity of $(80.0 \pm 16)$ km/s. Referring to this breakout velocity, we directly obtain the HD shock velocity of $(84 \pm 1.2)$ km/s, showing significant HD pressure boosting in the planar target. In addition, we also see that the strong HD shock caught up with the ID shock and suppressed it. These behaviors are similar to the hemispherical target but have no convergence effect.

With the DD laser of 3.6 kJ, the second shot with the 40 μm Al shield instead of the 30 μm Mo shield in the first shot gives the HD shock velocity similar to that in the review paper[18], thus the preliminary experimental result of HD pressure boosting is checked and further confirmed.

### HD pressure boosting and smoothing

With the help of numerical simulations by reproducing experimental data of the shocks in the quartz window we can get the HD pressure and density at the ablating CH surface for the hemispherical and planar targets. In view of the ablating CH surface expansion close to one-dimensional geometry, as explained in Methods, we use one-dimensional radiation hydrodynamic code RDMG[22] to simulate the implosion dynamics. Given the measured ID temperature history (Fig. 3a) and the DD laser power (Fig. 2b), only through tuning the flux limiter of the electron heat conduction to $f_e = 0.2$, the simulations reproduce the measured ID and DD shock velocity history and shock arriving time in the quartz very well, which is explained in the section of VISAR in Methods in detail, and are shown in Fig. 3c. We will discuss the results in the following.

### The smoothed maximal HD pressure in the hemispherical target

In the simulation matched with the experiment, we first mention the ID corona plasma background offered by the ID laser, where radiation temperature and the radiation ablation pressure have reached $T_r \approx 200$ eV and $P_a \approx 45$ Mbar, respectively, while pre-compressed the hemispherical target. At $t \sim 2.7$ ns, the ID shock driven by the radiation ablation pressure $P_a$ arrived in the quartz at point A with a simulation velocity of 55 km/cm (black) coinciding with the experimental value (red in Fig. 3c) very well. We now discuss HD pressure boosting and smoothing near the ablating CH surface. Thanks to the ID laser that offers a long-scale-length ID corona plasma, as shown in Fig. 4a. At

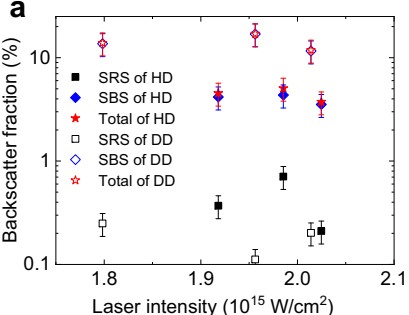
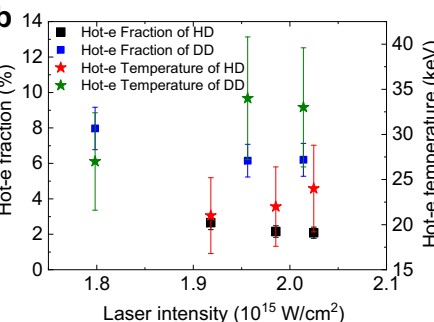
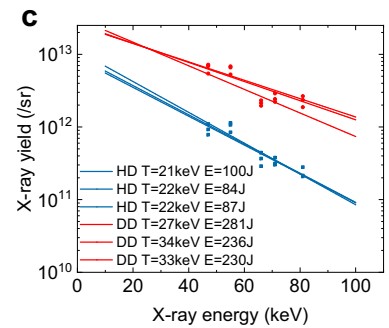

**Fig. 5 | Measurement of DD laser plasma interaction. a** Fraction of stimulated backscattering (SRS+SBS) vs laser intensity and the error bar reflecting the uncertainty in the experimental measurements; **b** hot-electron energy fraction (hot-e fraction) vs laser intensity and the error bar reflecting the uncertainty based on statistical uncertainty in the hard-X-ray data; **c** hot-electron temperature.

$t = 2.6$ ns, the supersonic-electronic-heat wave propagates from the critical surface at the radius $R \sim 770$ μm toward the radiation ablation front at $R \sim 530$ μm, and at $t \sim 3.02$ ns, it slows down to a compression wave following a precursor shock at the radius $R \sim 520$ μm, as seen in Fig. 4a, where $R = 0$ is in the center of the quartz sphere. This compression wave supported by 1-ns flattop pulse DD lasers, pushes low ID corona plasma density from continuously ablating CH ablator, forming an HD density plateau with a spatial width of about 19 μm between the radiation ablation front and compression wave front (two dotted lines in Fig. 4b), where the HD density and hence the HD pressure increases rapidly. At time $t \sim 3.7$ ns, coinciding with the VISAR time in the quartz, the strong HD shock with a velocity of ~93 km/s caught up with the relative-weak ID shock with a velocity of ~55 km/s, finally, the ID shock was suppressed, as seen in Fig. 3c. At that time, in the density plateau, electron temperature $T_e$, radiation temperature $T_r$, HD density, and HD pressure reach the maximum, as shown in Fig. 4b, c, where $T_e = T_r = 203$ eV, maximal HD density $\rho_{HD} \approx 1.36$ g/cm³, and the maximal HD pressure reaching $P_{HD} \approx \Gamma\rho_{HD}T_r \approx 180$ Mbar with experimental errors of ~± 4%. It is the plateau density that rose to make HD pressure boosting, resulting in $P_{HD}/P_a = \rho_{HD}/\rho_a \approx 4$. During the slowing down process, due to the perturbation wavelength $\lambda_p < 2\pi\beta\Delta R_{ID}$ the supersonic-electron-heat wave is thermally smoothed very well, where $\lambda_p \approx 500$ μm is equal to the FWHM of a Gaussian light spot, and for $T_r = 200$ eV, $\Delta R_{ID}$ equals about 240 μm and $\beta$ equals about 1.5. As a result, the boosted and smoothed HD pressure is achieved.

In the HD density plateau, from the measured HD pressure $P_{HD} \approx 180$ Mbar we acquire a corresponding HD density $\rho_{HD} \approx P_{HD}/(\Gamma T_r) \approx 1.36$ g/cm³, as seen in Fig. 4b, c, that satisfies the approximately fitted scaling relation of $\rho_{HD} \approx 0.92E_{DD}^{1/4}$, where the DD laser energy $E_{DD}$ in units of kJ.

In addition, for another round of the experiment for the ID laser energy of 48.5 kJ and the DD laser energy of 3.6 kJ with the measured radiation temperature of $T_r = (201 \pm 6)$ eV and HD shock velocity of $(82.7 \pm 2)$ km/s, we obtained the HD pressure $P_{HD} \approx 170$ Mbar, 3.8 times the radiation ablation pressure, and the inferred HD density of $\rho_{HD} \approx 1.28$ g/cm³.

### The smoothed maximal HD pressure in the planar target
From the simulations matched with the corresponding experimental data like in the hemispherical target, we, with the DD laser energy of 4.0 kJ, obtained the HD pressure of ~155 Mbar and a density of ~1.17 g/cm³. And, with the DD laser energy of 3.6 kJ, we obtained the HD pressure of ~150 Mbar and a density of ~1.15 g/cm³ in which the early experimental data in the ref. [18] was also checked.

### Fractions of stimulated backscattering and hot-electron energy
In addition to the HD pressure, we also measured the fractions of stimulated backscattering and hot-electron energy for the DD laser

intensity of ~1.8 PW/cm² which is needed for generating the supersonic-electronic-heat wave.

We first explain that the experiments on HD pressure boosting and smoothing are performed under conditions of the same radiation temperature of $T_r = (200 \pm 6)$ eV and the same DD laser energies of 3.6 and 4.0 kJ, and therefore, the hemispherical and the planar targets installed in the same semicylindrical hohlraum should have similar LPI. Here, we only discuss LPI occurring in the hemispherical ablator-semicylindrical hohlraum target.

Four DD laser beams propagate toward the critical surface at $R \sim 770$ μm in the semicylindrical hohlraum filled a $C_5H_{12}$ gas with a density of 1.2 mg/cm³, and are overlapped into a bundle near ~$n_c/4$ ($n_c \approx 10^{22}$ g/cm³ for the laser wavelength of 0.35 μm) at $R = \sim 1050$ μm. The backscattered laser beams go through FABS and NBS which are shown in Fig. 2a, to reach the diagnostic system, where laser backscattering energy and time-resolved spectrum are recorded. The measured total backscattering fractions involving stimulated Brillouin scattering (SBS) and stimulated Raman scattering (SRS) in HD are plotted in Fig. 5a, where the total averaged backscattering fraction (SBS+SRS) is ~$(4.5 \pm 0.7)$% (red stars) in the range of laser intensity $I_L = (1.8-2.2)$ PW/cm², and the backscattering fraction for SRS is only ~$(0.4 \pm 0.08)$% far less than that for SBS. However, in the traditional DD scheme, total averaged backscattering fraction (SBS + SRS) is as high as ~$(13 \pm 3.5)$% about 3 times that in HD, and SRS is smaller as seen in Fig. 5a. The experimental results for the lasers in the DD scheme are basically consistent with OMEGA-planar and OMEGA-spherical targets in the intensity range of $I_L \sim (1.5-2.8)$ PW/cm²[23,24].

We now discuss the hot electron energy fraction for the DD lasers in HD. Using FF spectrometer (Fig. 2a), the measured total averaged energy fractions are ~$(2 \pm 0.4)$% about 75 J only for the DD laser energy, and ~$(6.3 \pm 1)$% about ~220 J for the DD scheme, as seen in Fig. 5b. The hot-electron temperature from the measured X-ray spectrums is plotted in Fig. 5c, where the temperature of 21−22 keV (blue) is for the DD laser-plasma interaction in the HD scheme and of 27−34 keV (red) is for laser-plasma interaction in the DD scheme. These results show that due to the SRS fraction only ~$(0.4 \pm 0.08)$%, the hot-electron generation may mainly come from two-plasmon decay.

These data show that both the backscattering fraction and the hot-electron-energy fraction in the HD plasma that is in local thermodynamical equilibrium with temperature $T_r = T_e = T_i = 200$ eV are only about one-third of those in the DD scheme. So far, why the LPI in the HD scheme is smaller than that in the DD scheme that has no radiation ablation is still not very clear, which needs further exploration and discussion.

## Discussion
Our experimental results well demonstrate that the HD schemes can provide a well-smoothed HD pressure much larger than the radiation

**Table 1 | The simulation results of modeling ignition targets with spherical hohlraum of radius 5 mm**

| $R_c$ (μm) | $M_{DT}$ (mg) | $E_{ID}$ (kJ) | $T_r$ (eV) | $E_{DD}$ (kJ) | $\dot{W}_{DD}$ (TW) | $P_{HD}$ (Mbar) | $\rho_{HD}$ (g/cm³) | $V_{imp}$ (km/s) | $C_r$ | Yield (MJ) |
|---|---|---|---|---|---|---|---|---|---|---|
| 921 | 0.242 | 900 | 200 (290) | 875 | 350 | ~900 | ~4.90 | 425 | 23 | 17 |
| 925 | 0.245 | 900 | 200 (270) | 825 | 330 | ~820 | ~4.85 | 426 | 24 | 20 |
| 916 | 0.231 | 740 | 190 (270) | 750 | 300 | ~770 | ~4.60 | 415 | 24 | 14 |

ablation pressure. With the fitted scaling relationship between the HD pressure and laser energy, the scaled driving pressure for stable implosion and nonstagnation ignition agreed well with the error of about 15%, which provides an important reference to the high-gain ignition target design.

Using ID laser energy of $E_{ID}$ ~ 43–52 kJ, the peak radiation temperature of $T_r = 200 \pm 6$ eV is measured, as seen in Fig. 3a, which indicates that the semicylindrical hohlraum can provide an ID corona plasma environment to offer a large distance $\Delta R_{ID}$ for the "bulldozer" effect. It is also shown that according to the energy balance relationship, we can infer that with ID laser energy of 900 kJ, in the spherical hohlraum of a radius of 5 mm in the ignition target, the same radiation temperature of 200 eV can be achieved.

Using DD laser energy only $E_{DD} = 3.6$–4.0 kJ with the tuned intensities of ~1.8 PW/cm², for the measured radiation temperature of $T_r = 200$ eV we experimentally obtained the boosted HD pressures, for the hemispherical target $P_{HD} \approx 170$–180 Mbar about 3.8–4.0 times the radiation ablation pressure, corresponding to HD density $\rho_{HD} \approx 1.28$–1.36 g/cm³ in the density plateau, as seen in Fig. 4b, c, and for the planar target $P_{HD} \approx 150$–155 Mbar about 3.5–3.6 times the radiation ablation pressure, corresponding to HD density $\rho_{HD} \approx 1.15$–1.17 g/cm³, in which the previous result at 3.6 kJ in ref. 16 is checked and further confirmed. Thus, with more data from current and previous experiments, HD pressure boosting by the bulldozer effect is verified very well.

With the measured radiation temperature of $T_r = 200$ eV, which provides a large $\Delta R_{ID}$ ~ 240 μm, we demonstrate the HD pressure nonuniformity $\delta P/P < 1\%$, showing significantly thermal smoothing of the HD pressure.

The important phenomenon of the HD shock suppressing the asymmetric ID shock is observed experimentally, as seen in Fig. 3c, where no apparent remnant of any ID shock is seen near point B. It shows that in the layered DT capsule, the HD shock can suppress the asymmetric ID shock in the early implosion stage to prevent it from asymmetric implosion.

All the above experimental results confirmed the key effects in the HD scheme that would provide an effective path to stable implosion and high fusion energy for ICF.

We found from experimental data and the simulation results of the ignition targets that there is an approximately fitted hydroscaling relationship in the form $P_{HD}$ (Mbar) $\approx 180(E_{DD}/4)^{1/4}(T_r/2)$ and HD density $\rho_{HD} = P_{HD}/(\Gamma T_r) \approx 1.3(E_{DD}/4)^{1/4}$ (g/cm³). With this relationship, we can scale up 5–6 times the experimental data to obtain the maximal HD pressure and HD density for the ignition targets listed in the Methods section. The measured fractions of DD laser backscattering and hot electron energy for intensities of ~(1.5–2.8) PW/cm² are significantly lower than that in the DD scheme.

## Methods

### Simulation results for the modeling ignition targets

In Table 1, $R_c$ denotes the radius of the capsule, $M_{DT}$ is the DT mass. $T_r$ is the radiation temperature in the spherical hohlraum with a radius of 5 mm and the data in brackets denote peak radiation temperature (eV) higher than the hohlraum temperature $T_r$ because of the bremsstrahlung absorption in the optical-thick density plateau. $E_{ID}$, $E_{DD}$, $\dot{W}_{DD}$ denote ID and DD laser energy and DD laser power, respectively; $P_{HD}$ and $\rho_{HD}$ are the maximal HD pressure and HD density, respectively;

$V_{imp}$ is the maximal implosion velocity averaged by DT mass; $C_r$ is the convergence ratio in the ignition time; Yield denotes the released fusion energy.

For these modeling ignition targets, the adiabat $\alpha$ defined as the ratio of the DT pressure to the Fermi pressure is in the range of 3.2–3.4 at the maximal implosion velocity time.

### Experiments

The HD experiment was performed at SG-III where 20 ID laser beams irradiated the inner wall of a semicylindrical hohlraum and 4 delayed DD lasers were repointed to directly irradiate the radiation-ablated corona plasma. The hohlraum serves to convert the laser energy to soft X-rays, producing a temperature of 200 eV, together with the DD lasers of ~1.8 PW/cm² to hybridly drive the target at the bottom of the hohlraum. The scheme of the experiment is shown in Fig. 2.

### VISAR

Velocity Interferometer System with Any Reflector (VISAR) is used to measure shock velocity history in the present work. Probe lasers at 532 nm with a 10 ns pulse from VISAR are incident into the transparent material (quartz) through a mirror mounted below the quartz, as shown in Fig. 2c, and are reflected as the shock with the sharp front (density gradient) is coming to the quartz. The reflected probe lasers are collected by streak camera and recorded in the interferometer time-resolution imaging. The velocity history can be determined by calculating the fringes shifting caused by the Doppler effect and comparing it with the numerical simulations.

For this end, firstly, using the fringes shifting measured from the dual interferometer that has two etalons with thicknesses of 4 mm and 7 mm corresponding to the interference fringe sensitivities of 8.09 μm/ns/fringe and 4.62 μm/ns/fringe respectively, we obtained the sequence shock velocities spaced about 33 km/s. Then, under conditions of the ID laser energy of ~50 kJ and the DD laser energy of ~4 kJ, by tuning the flux limiter $f_e$, we achieve HD shock velocities by numerical simulation and compare them with the experimental data in sequence shock velocities. Because the HD shock velocity varying with $f_e$ can change only in a small range of far less than the velocity interval of 33 km/s, finally, we find the experimental HD shock velocity of ~93 km/s in the quartz at $f_e = 0.2$.

### Numerical simulations

In the experiment, the ID laser energy of ~50 kJ with pulse duration $\tau = 3$ ns absorbed in the gold inner wall of the semicylindrical hohlraum is converted into thermal X-rays with radiation temperature $T_r \approx 200$ eV. Meanwhile, the high temperature and pressure gold plasma moves radially inward, while the hemispherical CH plasma, which is ablated by the thermal X-ray, expands into the ID coronal plasma rapidly along the axial direction. The 2D simulation by LARED-series code showed that when the DD laser is unloaded the radial location of the quarter critical density of the gold plasma is at about 800 μm away from the hohlraum axis of the 2.5 mm diameter, and is far from the hemispherical CH plasma moving in the axial direction. This shows that there is no interaction between the gold plasma and CH plasma, and it is appropriate to use the one-dimensional radiation hydrodynamic code RDMG for re-emerging the experimental HD implosion dynamics. LARED-series include Euler and Lagrangian codes with the modules for fluid, multigroup-

diffusion or transfer treatment of radiation, and flux-limited electron and ion heat conduction. RDMG is a Lagrangian code, and the energy transport involves multigroup radiative transport and flux-limited electron and ion heat conduction.

## Data availability
The raw SG-III data are protected and are not available due to data privacy laws. Derived data supporting the findings of this study are available from the corresponding authors upon request.

## Code availability
The simulation codes used are not available to the general public due to intellectual property rights.

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

## Acknowledgements
Part of the work was supported by the National Natural Science Foundation of China under Grant Nos. 12075221 (J.Y.), 11975053 (L.W.), 12275032 (L.H.), 12235014 (J.Y.) and 12075220 (X.Z.), and the Science Challenge Project (TZJH1616-07) (J. Li). We thank Dr. Jing Wang and Dr. Zhu Lei for their help in plot. The simulations are carried out on the supercomputer in China.

## Author contributions

J.Y.: experimental arrangement, diagnostic plan and measurement synthesis; J. Li: target design and numerical simulations, computing analysis and data plot; X.T.H.: leading HD team, research idea, integrated physical analysis and paper writing; L.W.: implosion dynamics and hydrodynamic instabilities; Y.C.: simulation on thermal x-ray emission in spherical hohlraum; X.H.: calculation of plasma ionization degree; K.P.: experiment of laser-plasma interaction; F.W.: responsible for optic diagnosis; J. Liang: responsible for target fabrication; Z.G. and Y.L.: measurement of shock velocities by VISAR; X.L.: laser-plasma interaction diagnosis; X.C.: measuring radiation temperature; Z.C.: assistant for experimental arrangement; X.Z.: working in X-ray imaging diagnosis; Y.X.: for analysis on DD laser nonuniformity; B.L., M.H., H.C., L.H., Z.L., and C.Z.: for analysis and simulation on DD laser–plasma interaction; Z.D. and Z.F.: participating in implosion simulations; B.Q.: participating in physical discussion; F.L.: laser beam arrangement; S.J.: responsible for overall experiment coordination; M.Y.Y.: taking part in writing; S.Z.: participating in physical analysis.

## Competing interests
The authors declare no competing interests.
