## [Peer Review File · Nature Communications]

Experimental confirmation of driving pressure boosting and smoothing for hybrid-drive inertial fusion at the 100-kJ laser facilityREVIEWER COMMENTS

Reviewer #1 (Remarks to the Author):

I believe that this paper is very interesting and it is an innovative and substantial contribution to the field of Inertial Confinement Fusion. Publication of this article would be very timely seen the recent breakthrough in laser fusion obtained at NIF.

In general I have only a few remarks which I would ask the authors to reply before publication.

1) in the conclusions the authors write that "Using DD laser with intensities $\sim (1.8 - 2.0)$ PW/cm², ... we stably obtained the boosted HD pressure $P \approx 150 - 155$ Mbar about 3.5-3.6 times the radiation ablation pressure experimentally"

Well, this indeed the range of pressure which can be obtained in direct-drive or similar intensities, using e.g. Lindl's scaling. So in this case the contribution from the indirect drive part of the scheme does not seem to be really apparent. Can the authors comment on this?

2) While the comparison between the HD scheme and the "traditional" DD scheme is well described, I still have doubts on how the HD scheme compares to the Shock Ignition approach. SI is also based on a two step process with a first "compression" laser pulse and a final "ignition" laser spike.

The authors write that "the distance between the electron ablation front and critical surface for the traditional DD system is too short". Does this conclusion also apply to shock ignition?

3) Does this conclusion also apply to the proposed scheme of "foam buffered target" in which the pellet is surrounded by a low density material (foam) to create a long scale length plasma assumed to be able to smooth the non-uniformities thanks to the long stand-off distance available to thermal smoothing?

4) what is the measured "conversion efficiency for thermal X-rays" reported at the end of page 5 (section "Experimental target design")

5) the authors speak about a "hot electron energy fraction of the DD lasers in laser-plasma interaction in HD..." of about 2% and they say that this is "significantly lower than that in the traditional DD laser". However I found no measurement or estimation of the HE temperature. This is indeed an additional important parameters because if the HE are not too energetic, they will not be able to penetrate deeply into the fuel, and so their harmful effect will be strongly mitigated.

6) Also, what is the fraction of HE produced during the ID phase?

7) The authors write that SRS amounts to about 0.5 %. Since in these experimental conditions SRS is probably the main source of HE, how is this result compatible with a HE conversion efficiency of 2% ??

Reviewer #2 (Remarks to the Author):

This manuscript reports on simulation and experimental work on the hybrid-drive (HD) approach to inertial confinement fusion (ICF). This approach uses a spherical hohlraum and long-pulse laser drive to provide an 'indirect' X-ray drive to the ICF capsule, combined with a short pulse indirect laser drive applied sometime after the start of the long pulse. If tuned correctly, the relativistic electron heat wave generated by the short pulse laser slows to form a compression wave in the material ablated by the indirect drive. The net effect is a stable plateau in density between the indirect-drive ablation front and direct-drive critical surface, increasing the drive pressure on the ICF capsule and thermally smoothing perturbations. It is hoped that his hybrid approach will provide favorable scaling to ignition conditions and an effective path to high fusion yield from ICF implosions.

Inertial fusion experiments have been widely reported recently as a result of significant advances in a related approach using only a direct drive. This makes ICF experiments, and alternative paths to ignition, a timely contribution and of high interest to a wide audience. However, the work presented here is an incremental change to work that has already been reported; the HD approach has been widely described over several decades, and nearly identical simulations and experiments have been published recently (High Energy Density Physics 96, 100804). The only difference between this submission and the previously reported results is a slight increase in direct drive energy (4.0 KJ vs 3.6KJ); the key results – observation of pressure boosting, peak radiation temperature, and laser backscatter - are unchanged.

Given the incremental nature of this work, I do not find it suitable for publishing in Nature Communications. However, the topic of hybrid-drive, and new experimental results for ICF, remains highly impactful and relevant to the Nature readership. I therefore recommend that the authors re-write the manuscript to avoid repeating previously published work, and to identify the novel and high-impact results of the present work. I also suggest that the authors pay attention to the clarity and conciseness of the new manuscript, since I found the current submission difficult to read and somewhat unclear in some places.

Reviewer #3 (Remarks to the Author):

Dear Editor,

This manuscript reports on initial experiments at SG-III using a hybrid-drive (HD) scheme designed to take advantage of the drive smoothing provided by indirect-drive (ID), and the efficient absorption of laser energy provided by direct-drive (DD). The manuscript presents important proof-of-principle experiments for the HD scheme that can be used to validate modeling capabilities and assess the LPI threat. The manuscript is well organized and provides an appropriate level of detail and could be appropriate for publication with some changes. My specific comments follow:

- 1) Although the manuscript is well-written from the point-of-view of the results being presented in a logical and coherent manner, there are a large number of grammatical errors that make it difficult to read.
- 2) The abstract (and elsewhere) states that the HD shock “stops the asymmetric ID shock.” I am not sure what this statement is supposed to mean, but certainly it is not meant to be taken literally.
- 3) As the HD scheme is proposed as an alternative to DD/ID, it would help if the manuscript was more explicit about the advantages/disadvantages of the HD scheme relative to those schemes. For example, the HD scheme provides increased ablation pressure relative to ID, but how does the pressure compare to DD alone?
- 4) In the caption of Fig. 1 LHEs should be LEHs.
- 5) Figure 2(c) shows an Al layer, but in the text this is referred to as a Mo layer (the figure caption says that it can be Mo or Al).
- 6) The manuscript states that the measured radiation temperatures agreed with predictions. It would be nice to include the predicted radiation temperature in Fig. 3(a).
- 7) What does the colormap represent in Fig. 4(a)?

8) In the first sentence on page 9 “reappearing” should be “reproducing”.

9) In the 2nd paragraph on page 11 “75J for DD” should be “75J for HD”.

10) The manuscript states in regards to the reduction in hot-electron fraction and backscatter fraction for the HD scheme: “Obviously, the thermal smoothing effect in HD significantly reduces the fractions compared with the traditional DD.” This is not at all obvious to me. Traditionally I think of thermal smoothing as mitigating hydrodynamic instabilities, not laser plasma instabilities. The authors should explain how thermal smoothing is mitigating LPI in this case.

11) Despite the mitigation relative to DD, the backscatter and hot-electron production in the HD scheme is still quite high and is likely to get worse when scaling up to a high-gain design. Are these levels believed to be acceptable for the HD design or are there plans for mitigation?

12) What is the f-number of the drive beams, as this has a significant impact on SBS/SRS backscatter?

Response Letter

Article ID: NCOMMS-22-52906

Title: Experimental confirmation of drive pressure boosting and smoothing for hybrid-drive inertial fusion at the 100-kJ laser facility

Authors: Ji Yan, Jiwei Li, X. T. He, et al.

We are grateful to the three referees for their insightful reviews and valuable comments on our manuscript. According to their comments, we have carefully revised the manuscript. Following are the specific responses to the referees' comments point by point.

I. Responses to Referee #1

1. Comment: *"in the conclusions the authors wriest that "Using DD laser with intensities $\sim (1.8 - 2.0)$ PW/cm², ..., we stably obtained the boosted HD pressure $P \approx 150 - 155$ Mbar about 3.5-3.6 times the radiation ablation pressure experimentally." Well, this indeed the range of pressure which can be obtained in direct-drive or similar intensities, using e.g. Lindl's scaling. So in this case the contribution from the indirect drive part of the scheme does not seem to be really apparent. Can the authors comment on this?"*

Response:

We understand the reviewer's concern. Yes, under the current laser drive condition of the direct-drive (DD) laser energies of 3.6-4 kJ only, the hybrid-drive (HD) pressure of 150-155 Mbar in our HD scheme is similar to that in the DD scheme. However, the physics is completely different. In our HD scheme, the important roles of the indirect drive (ID) lasers are to produce an ID corona plasma background and a large enough distance ΔR_{ID} between the critical surface and radiation ablation front before the DD laser arriving, and also a radiation ablation pressure at the radiation ablation front in addition to pre-compression of the fuel. Afterwards, the DD laser with the intensities of $I_L \sim (1.5-2.0)$ PW/cm² then is absorbed near the critical surface and converted into a supersonic-electron-heat-conduction wave (below, abbreviated the SEC wave). The SEC wave propagates within the large ΔR_{ID} from the critical surface toward the radiation ablation front and slows down to a plasma compressive wave while smoothing. This compressive wave isothermally compresses low ID corona plasma density ρ_{ID} into high HD density ρ_{HD} , which can be fitted as the function of the ID laser energy $\sim E_{DD}^{1/4}$ (see the manuscript), to change the radiation ablation pressure $P_{ID} = \Gamma \rho_{ID} T_r$ at the radiation ablation front into a new smoothed HD pressure $P_{HD} = \Gamma \rho_{HD} T_r \propto E_{DD}^{1/4} T_r$, where E_{DD} is DD laser energy and T_r is radiation temperature. Therefore, we see that the first ID process plays a key role in our HD scheme, which results in that the HD pressure scales as both the DD laser energy $E_{DD}^{1/4}$ and the radiation temperature, while in the DD scheme, the pressure only scales as $I_L^{2/3}$.

For the CH ablator, the HD pressure achieved in our HD scheme can be calculated as $P_{HD}(\text{Mbar}) \approx 62 E_{DD}^{1/4} T_r$, while the DD pressure achieved in the DD scheme (without ID) is $P_{DD} \approx 90 I_L^{2/3}$. Under ~ current laser conditions in our experiments, the DD laser energy is $E_{DD} = 3.6 - 4$ kJ, $I_L = 1.8$ PW/cm², and $T_r = 200$ eV, the pressure in our HD scheme is about $P_{HD} \sim 155$ Mbar, which is indeed close to that in the DD scheme $P_{DD} \sim 140$ Mbar. However, when we scale up to the ignition condition with the increased DD laser energy, such as $E_{DD} = 825$ kJ, the pressure in our HD scheme can be much boosted (due to the role of the ID) up to $P_{HD} \sim 775$ Mbar, while that in the DD scheme still keeps to be as only 140 Mbar, since

the laser intensity is similar as $I_L = 1.8 \text{ PW/cm}^2$.

In the present revised manuscript, we also added a hemispherical ablator target performed in a recent experiment and under the same DD laser energy of 3.6 - 4.0 kJ and obtained the HD pressure $P_{HD} \sim 170 - 180 \text{ Mbar}$ higher than $\sim 150 - 155 \text{ Mar}$ in the planar ablator target. These results completely verified the HD pressure boosting and smoothing effects-the heart of the HD scheme

2. Comment: *“While the comparison between the HD scheme and the “traditional” DD scheme is well described, I still have doubts on how the HD scheme compares to the Shock Ignition approach. SI is also based on a two-step process with a first “compression” laser pulse and a final “ignition” laser spike. The authors write that “the distance between the electron ablation front and critical surface for the traditional DD system is too short”. Does this conclusion also apply to shock ignition?”*

Response:

The two-step process of ID and DD in our HD scheme is inherently different from the “compression” and “ignition” in the DD scheme. This conclusion also applies to SI which is under the DD scheme. As explained in the manuscript, in HD, in the first step, the ID distance ΔR_{ID} offered in advance by the ID laser is about 250-300 μm for radiation temperature $T_r = 200 \text{ eV}$, which is large enough for the SEC wave to slow down to a compressive wave thermally compressing the ID corona plasma density into high plasma density to make the HD pressure far higher than the radiation ablation pressure. However, in SI, no ID laser is used in advance, and thus of course there exists no large ID distance ΔR_{ID} before the DD laser arrives at the critical surface. Although in the later stage of SI, the DD distance ΔR_{DD} between the critical surface and electron ablation front offered by the pre-pulse DD laser ablating the ablator surface is getting to increase, about 100-150 μm for a model in Fig. 1 in Nucl. Fusion 54, 054004(2014), due to the mass ablation rate for electrons being smaller than for radiation, under the same laser intensity ΔR_{DD} is shorter than ΔR_{ID} , resulting in the SEC wave driven by the DD laser directly hitting the capsule without enough time slowing down to a compressive wave. Therefore, the electron ablation pressure $P_{DD} \sim I_L^{2/3}$ can't be smoothed very well, which leads to an asymmetric and unstable implosion in the early stage of the implosion. In addition, in SI, the driving pressure (electron ablation pressure) P_{DD} over 300 Mbar is required for the hotspot ignition (Nucl. Fusion 54, 054004(2014)) in which the DD laser intensity of $I_L \sim 5 \text{ PW/cm}^2$ is required. For such intensity, the velocity of the SEC wave slowing down to the compressive wave should be larger than that for $I_L = 1.8 \text{ PW/cm}^2$ in HD, and therefore, the DD distance ΔR_{DD} has too far larger than 250-300 μm .

3. Comment: *“Does this conclusion also apply to the proposed scheme of “foam buffered target” in which the pellet is surrounded by a low-density material (foam) to create a long scale length plasma assumed to be able to smooth the non-uniformities thanks to the long stand-off distance available to thermal smoothing?”*

Response:

In HD, the distance ΔR_{ID} offered in advance by the ID laser does at least two things, that is, the SEC wave will be slowed down to a compressive wave for boosting the HD pressure and thermally smoothed very well.

To our understanding, the foam (low density but above critical density) indeed removes the critical surface of the ablator in the pellet to the foam surface and is available for thermal smoothing well in the form $\exp(-\ell\Delta R/R_{cr})$ if the foam thickness ΔR is larger than the perturbation wavelength $\lambda = 2\pi R_{cr}/\ell$

from the DD laser imprinting and beam overlapping, ℓ is the perturbative mode number, and R_{cR} is the radius of the critical surface.

However, the foam is difficult to offer a large distance, like ΔR_{ID} in HD, since it will significantly reduce laser energy to the pellet.

4. Comment: “*what is the measured "conversion efficiency for thermal X-rays" reported at the end of page 5 (section "Experimental target design")*”

Response:

Thanks for the reviewer’s correction. It refers to the conversion efficiency η of the ID laser to thermal X-rays. The x-ray conversion efficiency is estimated based on the hohlraum power balance equation. The equation can be described as follow:

$$\eta(P_{laser} - P_{backscatter}) = [A_{wall}(1 - \alpha) + A_{LEH} + F_{CH}A_{CH}] \sigma T_r^4$$

where P_{laser} and $P_{backscatter}$ are the incident laser power and the backscattered laser power due to LPI effect, respectively. A_{wall} , A_{LEH} and A_{CH} represent the surface areas of the hohlraum wall, laser entrance hole and CH sample, respectively. α is the albedo of the Hohlraum and inferred by analytical model ($\alpha = 1 - 0.32T(100eV)_r^{-0.7} \tau^{-0.382}$). F_{CH} represents the ratio of the absorbed flux over the incident flux of capsule and approximately as a constant and equal to 0.7. σ is the Stefan-Boltzmann constant. T_r is the radiation temperature which measured by calibrated flat response x-ray diode (FXRD). The uncertainty of radiation temperature using FXRD is 3%. In our experiments, the conversion efficiency is $89 \pm 8\%$.

5. Comment: “*the authors speak about a "hot electron energy fraction of the DD lasers in laser-plasma interaction in HD..." of about 2% and they say that this is "significantly lower than that in the traditional DD laser". However, I found no measurement or estimation of the HE temperature. This is indeed an additional important parameter because if the HE are not too energetic, they will not be able to penetrate deeply into the fuel, and so their armful effect will be strongly mitigated.*”

Response:

Thanks for the reviewer’s comments, we have added a diagram of the measured HE temperature in the present revised manuscript (see Fig. 5c). In our experiment, the HE temperature is about 21-22 keV. The production of HE is mainly from two-plasmon decay (TPD) since the backscattering fraction for SRS is only $\sim 0.4\%$ experimentally.

The simulation shows that for the modeling ignition target, the area density of the remainder of the ablating CH ablator is about 0.06 g/cm^2 at the time of the maximal implosion velocity in which 0.02 g/cm^2 is in the density plateau piled up by the isothermal compression. On the other hand, the HE’s range can be expressed in the form $\rho R(\text{g/cm}^2) \approx 0.59 \times 10^{-5} E_e^{1.661}$ for $E_e > 2 \text{ keV}$ in the CH material (Phys. Plasmas 18, 022703(2011)), where E_e is the HE energy in keV. We see from the expression and numerical result that all hot electrons with energy less than 250 keV are completely trapped in the remaining CH, impossible to reach the fuel. Obviously, for the HE temperature of 21-22 keV the number of the HEs with energy greater than 250 keV can be neglectable and they are impossible to preheat the fuel significantly.

6. Comment: “*Also, what is the fraction of HE produced during the ID phase?*”

Response:

A calibrated filtered-Fluorescence (FF) hard x-ray spectrometer was employed to monitor the fraction of the hot-electrons. In HD shots, the measured hot-electrons include both ID phase and DD phase. But however, in ID only shots, the measured energy of hot-electrons is lower than 5 J (the detectable threshold of the spectrometer) and the fraction of HE produced during the ID phase is around $5 \text{ J}/43 \text{ kJ}=0.01\%$, since the ID laser energy is 43kJ in our experiment.

In the HD experiment, the ID laser with an averaged intensity of about 0.4 PW/cm^2 and energy of 40-50 kJ is coupled in the HD plasma with the DD laser with an intensity of $1.5\text{-}2 \text{ PW/cm}^2$ and energy of 3.6-4.0 kJ launching in a later pulse duration of the ID laser, and therefore, it is difficult to separate the HE energy from the ID laser and DD laser. We have measured HE produced by the ID laser energy of 43 kJ without the DD laser, and its energy is about 3 J, which means that the energy fraction of HE produced by the ID laser is $3\text{J}/43\text{kJ} = 0.0067\%$.

7. Comment: *“The authors write that SRS amounts to about 0.5 %. Since in these experimental conditions, SRS is probably the main source of HE, how is this result compatible with a HE conversion efficiency of 2%?”*

Response:

There should be some misunderstanding. As mentioned in our manuscript, in our HD scheme, as the laser intensity $I_L \sim 1.8 \text{ PW/cm}^2$ exceeds the threshold of two-plasma decay (TPD), TPD, instead of SRS, is the main source of HE. That is the reason why SRS amounts to only about 0.5%. The main component of HE (about 2%) comes from TPD. In the HD experiment with the DD laser intensity of $\sim 1.8 \text{ PW/cm}^2$, we measured the SRS fraction of about 0.4 % on SG-III and compared it to the results on OMEGA under the DD scheme with the same intensity, as seen in Fig. 20 in the reference No. 23 and in Fig. 21 in the reference No. 24 in our manuscript., where the SRS fraction is about 2.5-3 times the fraction of 0.4% in the HD experiment and SBS dominated. Considering that the radiation ablation offered a uniform ID corona plasma before the DD laser arrives, different from the DD laser-plasma interaction (LPI) in the DD scheme, the thermal smoothing effect on the ID corona plasma may result in the decrease of the SRS fraction in the HD experiment. So, the SRS fraction on SG-III may be reasonably compared with the results in OMEGA. Such a low SRS fraction in HD is unable to offer the HE energy fraction of $\sim 2\%$ for the DD laser energy of 3.6-4.0 kJ. Therefore, we think the HE fraction mainly comes from two-plasmon decay (TPD) since this intensity of $I_L \sim 1.8 \text{ PW/cm}^2$ exceeds the threshold of TPD.

At electron temperature of $T_e = 2 - 3 \text{ keV}$ and the electron number density of $n_e/n_c \sim 0.2 - 0.7$, the maximal linear growth rate for TPD is $\sim \gamma/\omega \sim 10^{-2}/(n_e/n_c)$, where n_c the critical number density, the laser frequency $\omega_0 \approx 5.4 \times 10^{15}/\text{s}$ for the wavelength of $0.35 \text{ }\mu\text{m}$. Therefore, at a quarter of the critical density, we have the maximal linear growth rate of $\gamma \sim 0.2/\text{fs}$, and the occurrence of TPD is completely possible soon after the DD laser arrives. Further specialized investigation of course is necessary.

II. Responses to Referee #2

1. Comment: *“Inertial fusion experiments have been widely reported recently as a result of significant advances in a related approach using only a direct drive. This makes ICF experiments, and alternative paths to ignition, a timely contribution and of high interest to a wide audience. However, the work presented here is an incremental change to work that has already been reported; the HD approach has been widely*

described over several decades, and nearly identical simulations and experiments have been published recently (High Energy Density Physics 96, 100804). The only difference between this submission and the previously reported results is a slight increase in direct drive energy (4.0 KJ vs 3.6KJ); the key results – observation of pressure boosting, peak radiation temperature, and laser backscatter - are unchanged.”

Response:

Thanks for the referee’s comments. We now explained those as follows.

The hybrid-drive content is only a portion of the conference proceedings review paper issued in HEDP, where we simply presented with DD laser energy of 3.6 kJ one early experiment data of the HD pressure of ~ 150 Mbar without more direct experimental evidence and the detailed analyses to confirm the reliability of HD pressure boosting and smoothing, which are the key effects for our HD scheme.

In our previous manuscript, we analyzed new and published experimental data in detail to demonstrate and confirm the HD pressure boosting and smoothing effects.

Firstly, with the new DD laser energy of 4.0 kJ and a Mo shield instead of Al, we obtained the new experimental result of the HD pressure of ~ 155 Mbar, and with DD laser energy of 3.6 kJ checked the result of ~ 150 Mbar in the paper issued in HEDP. Thus, we further confirmed the HD pressure-boosting effect physically with more experimental data. In addition, we first showed HD pressure smoothing and observed the phenomenon of the HD shock merged with the ID shock (Fig. 3c), both are important for stable implosion. Second, we showed how under keeping the radiation temperature of 200 eV unchanged the experimental target is designed by scaling down the size of the spherical hohlraum in the ignition target, which makes the experimental results can directly scale up to the ignition target experimentally. Finally, in the manuscript, we found that with a pre-offered ID corona plasma background of radiation temperature 200 eV, tuning the distance between the critical surface and the radiation front and the slowing down length, the supersonic-electronic-heat wave converted by the DD laser intensity of 1.8 PW/cm^2 can provide a perfect bulldozer effect (a compressive wave) to generate significant HD pressure boosting and smoothing effect. we further discussed the time evolution process of indirect-driven and hybrid-driven shock velocities experimentally for the planar target and deduced the boosted and smoothed pressure in the density plateau by the simulations matched with experimental results in the quartz. This makes the manuscript a new paper essentially different from the paper in HEDP.

As for why the key results of peak radiation temperature and laser backscatter are unchanged. Because the peak radiation temperature of 200 eV is the optimal temperature in the design of the ignition target, which can well provide the conditions of HD pressure boosting and smoothing, and also the right ID energy for the ignition target, in addition, according to the indirect-drive energy balance relationship, when the semicylindrical hohlraum used in the experiments is designed by scaling down the spherical hohlraum in the ignition target, the peak radiation temperature of $T_r=200$ eV is the same as in the ignition target, which is beneficial to scaling up the experimental results to the ignition target. This is why peak radiation temperature $T_r=200$ eV is unchanged in the review article in HEDP and the manuscript. In addition, the same radiation temperature heating the surface of the CH ablator leads to the same plasma environment for LPI, and therefore, for the DD laser intensity of $\sim 1.8 \text{ PW/cm}^2$, the backscatter and the hot-electro energy should be close to those in similar experiments, which is plotted in Fig. 5c.

2. Comment: *“Given the incremental nature of this work, I do not find it suitable for publishing in Nature Communications. However, the topic of hybrid-drive, and new experimental results for ICF, remains highly*

impactful and relevant to the Nature readership. I therefore recommend that the authors re-write the manuscript to avoid repeating previously published work, and to identify the novel and high-impact results of the present work. I also suggest that the authors pay attention to the clarity and conciseness of the new manuscript, since I found the current submission difficult to read and somewhat unclear in some places.”

Response:

Although we think our previous manuscript is essentially different from the paper issued in HEDP, we, to respond to the reviewer’s comment of “to avoid repeating previously published work, and to identify the novel and high-impact results of the present work”, in rewriting the manuscript, added new experimental results with a hemispherical ablator target recently performed on SG-III. Under the same DD laser energies of ~ 3.6 kJ and ~ 4.0 kJ and radiation temperature of 200 eV, the hemispherical target due to the spherical convergence effect provided the peak HD pressure achieved 170-180 Mbar, larger than 150-155 Mbar of the planar target, as seen in Fig. 3c for the HD shock velocity and in Fig. 4c for the HD pressure of 180 Mbar. These results are the latest and greatest for the HD pressure experimentally to date and we discussed them in the text in detail.

We also tried to make some changes in the text to make the reading clear.

III. Responses to Referee #3

1. Comment: *“Although the manuscript is well-written from the point-of-view of the results being presented in a logical and coherent manner, there are a large number of grammatical errors that make it difficult to read.”*

Response:

Sorry, we have made an effort to correct some grammatical errors in the present revised manuscript.

2. Comment: *“The abstract (and elsewhere) states that the HD shock “stops the asymmetric ID shock.” I am not sure what this statement is supposed to mean, but certainly it is not meant to be taken literally.”*

Response:

We wrote it too simply in the abstract, more details are below:

A strong symmetric HD shock driven by the ideal HD pressure rapidly entering the imploding capsule collides in the opposite directions with the asymmetric relatively weak ID shock which is reflected from the center of the hotspot after pre-compressed the fuel and just arriving at the interface of the hotspot, and the ID shock reflected inward becomes weaker and is quickly caught up, swallowed and merged by the strong HD shock. Thus, the asymmetric ID shock in the early implosion stage is suppressed to prevent it from further asymmetric implosion.

We have made corresponding changes in the manuscript.

3. Comment: *“As the HD scheme is proposed as an alternative to DD/ID, it would help if the manuscript was more explicit about the advantages/disadvantages of the HD scheme relative to those schemes. For example, the HD scheme provides increased ablation pressure relative to ID, but how does the pressure compare to DD alone?”*

Response:

In the DD scheme, its advantage is having a high conversion efficiency of laser energy to the capsule, however, due to the mass ablation rate for electrons smaller than that for radiation the electron-conduction

region between the critical surface and the electron ablation front is relatively narrow, which easily transfers the nonuniformities from laser imprinting and beam overlapping to the imploding capsule, especially in the early stage, resulting in hydrodynamic instabilities. In addition, at the electron ablation front, the electron ablation pressure driving implosion is $P_{DD} = \Gamma \rho_{DD} T_e$, where Γ is a pressure constant for an ideal gas, the electron corona density ρ_{DD} is approximately $2\rho_c$ with ρ_c the critical density, and the electron temperature is $T_e \propto I_L^{2/3}$ where I_L the DD laser intensity (Ref. 9 in the manuscript). For the CH ablator, the critical density is $\rho_c \approx 0.03$ g/cc, and therefore, P_{DD} (Mbar) $\approx 90 \times I_L^{2/3}$ for the laser wavelength of 0.35 micron, where I_L in units of PW/cm². It is seen from the above discussion that due to the expansion of the ablator surface by high-temperature ablating, the plasma density ρ_{DD} at the electron ablation front is low compared with the normal ablator density. Therefore, in order to boost the DD pressure, the only way is to increase the electron temperature T_e or the DD laser intensity I_L . As an example, In the DD scheme, if the pressure driving the implosion is required to be $P_{DD} \approx 300$ Mbar (Batani et al., Nucl. Fusion 54, 2014), the laser intensity I_L must be over 5 PW/cm², resulting in severe LPI. In addition, laser imprinting and beam overlapping would lead to asymmetric implosion and hydrodynamic instabilities.

As for HD, a coupling of ID and DD, we use the advantages of ID (large mass ablation rate for radiation) and DD (high efficiency) to improve their shortcomings involving hydrodynamic instabilities. As we explained in the manuscript that before the DD laser arrives, the ID laser has offered a large distance ΔR_{ID} between the critical surface and the radiation ablation front, and then a supersonic-electron-heat wave converted by the DD laser with the intensity of $I_L = (1 - 2)$ PW/cm² propagates in ΔR_{ID} and slows down to a compressive wave before reaching the radiation ablation front, while thermal smoothing. This compressive wave, like a “bulldoze, isothermally compresses low ID corona plasma density ρ_{ID} , between the compressive wave front and the radiation ablation front, into high HD plasma density ρ_{HD} (fitted to $\sim E_{DD}^{1/4}$, see the manuscript) far greater than ρ_{ID} , and therefore, by increasing the plasma density rather than radiation temperature T_r , the radiation ablation pressure is changed into a new smoothed HD pressure, much greater than the radiation ablation pressure, i.e., $P_{HD}/P_{ID} = \rho_{HD}/\rho_{ID} \gg 1$. Clearly, the HD pressure increases with the DD laser energy $E_{DD}^{1/4}$ and radiation temperature T_r and is independent of the laser intensity. As an example, for the modeling ignition target with the radius 5 mm of the spherical hohlraum and radiation temperature $T_r = 200$ eV, at the radiation ablation front the HD maximal pressure P_{HD} can reach as high as 775 Mbar for $E_{DD} = 825$ kJ while the radiation ablation pressure is only $P_{ID} \sim 43$ Mbar, and at the electron ablation front for the intensities of $I_L = (1 - 2)$ PW/cm² the DD pressure is $P_{DD} \sim 90 - 158$ Mbar.

As shown in the experiment, with the target consisting of the semicylindrical hohlraum and the planar ablator, for the radiation temperature $T_r = 200$ eV, only using the low DD laser energy of 3.6 – 4 kJ, we obtain the HD pressure of 150-155 Mbar, about 3.5 times the radiation ablation pressure.

In the present revised manuscript, we added the new experimental results by using a new target consisting of the same semicylindrical hohlraum and a new hemispherical ablator. Using the same DD laser energy of 3.5-4.0 kJ and the same radiation temperature of 200 eV, the new experimental results, due to the spherical convergent effect, achieve HD pressures of 170-180 Mbar higher than the HD pressures of 152-155 Mbar in the planar target. These results further verified the HD pressure boosting and smoothing effects-the heart of the HD scheme and are discussed in detail in the revised manuscript.

4. Comment: *“In the caption of Fig. 1 LHEs should be LEHs.”*

Response:

Thanks for the reviewer's correction, we have revised it in the present manuscript.

5. Comment: *“Figure 2(c) shows an Al layer, but in the text this is referred to as a Mo layer (the figure caption says that it can be Mo or Al).”*

Response:

Sorry, we were careless. In the last manuscript, we did two rounds of experimental shots. In the first shot, with the DD laser energy of 3.6 kJ, Al was used as the shielding layer as plotted in Figure 2(c), and in the second shot with the DD laser energy of 4.0 kJ, the shielding layer was Mo. In the discussion, only Mo was used.

In the new experiments, with the DD laser energies of ~ 3.6 kJ and 4.0 kJ the shielding layers for two shots all are Al, as shown in new Fig. 2(c).

6. Comment: *“The manuscript states that the measured radiation temperatures agreed with predictions. It would be nice to include the predicted radiation temperature in Fig. 3(a).”*

Response:

Following the reviewer's suggestion, we have included the predicted radiation temperature in Fig. 3(a). After several experimental checks, the simulation results of temperature are consistent with the experimental results within the experimental error range of $\pm 3\%$.

7. Comment: *“What does the colormap represent in Fig. 4(a)?”*

Response:

The simulated density versus the time and space.

8. Comment: *“In the first sentence on page 9 “reappearing” should be “reproducing”.”*

Response:

Thanks for the reviewer's correction, we have revised it in the new manuscript.

9. Comment: *“In the 2nd paragraph on page 11 “75J for DD” should be “75J for HD”.”*

Response:

Sorry for the unclear statement. In the previous manuscript, the total laser energies of ~ 47 kJ including 43 kJ for ID and 3.6-4.0 kJ for DD. Only 75J is for DD lasers, covering only a small portion. We have revised the texts.

10. Comment: *“The manuscript states in regards to the reduction in hot-electron fraction and backscatter fraction for the HD scheme: “Obviously, the thermal smoothing effect in HD significantly reduces the fractions compared with the traditional DD.” This is not at all obvious to me. Traditionally I think of thermal smoothing as mitigating hydrodynamic instabilities, not laser plasma instabilities. The authors should explain how thermal smoothing is mitigating LPI in this case.”*

Response:

In the DD scheme, when the DD lasers are incident on the ablator surface, especially in the early incident stage, it is inevitable that nonuniformities, like small density-depletion hollows or density bulges, appear in

the rapidly expanding electron corona plasma. These nonuniformities are difficult to thermally smooth out due to the low mass ablation rate for electrons.

As an example, we discuss the self-focus and filamentation in these small hollows when the DD laser with the intensity $I_L = 1.8 \text{ PW/cm}^2$ is incident on a quarter critical density surface with the plasma density of $\rho \sim 0.0075/\text{cc}$ for the CH ablator, in which the “collision” doesn’t dominate. In this case, the laser is soon focused into the hollows unstably due to the light refraction index increasing caused by electron number density depletion. For the DD laser intensity of $I_L = 1.8 \text{ PW/cm}^2$, electron temperature T_e can rise to $\sim 4.5 \text{ keV}$ soon through inverse bremsstrahlung absorption while the ion temperature rises to $T_i \sim 1.0 \text{ keV}$ by the time at least $\sim 1 \text{ ns}$ (refer to ref. 9 in the manuscript). At the quarter critical surface, the thermal pressure of electrons is $n_e T_e \sim 1.4 \times 10^6 \text{ J/cc}$ while the laser radiation pressure is $I_L/c \sim 0.6 \times 10^5 \text{ J/cc}$, where c light speed. Thus, if n_e is drained by 10%, due to the pressure balance the laser intensity must be increased to $\sim 4.1 \text{ PW/cm}^2$, which results in severe LPI.

In HD, as discussed in the manuscript, the ID laser provided the ID corona CH plasma (completely ionized) in local thermodynamic equilibrium with peak temperature $T_r = T_e = T_i = 200 \text{ eV}$ within several nanoseconds before the arrival of the DD laser with the intensity of 1.8 pw/cm^2 for the wavelength of $0.35 \mu\text{m}$, where T_r , T_e , T_i the temperatures for radiation (thermal X-rays), electron, and ion, respectively. Such LTE plasma offers a radiation-hydrodynamic sound velocity of $C_T = 110 \mu\text{m/ns}$, the density nonuniformity around hollows or bulges can be thermally smoothed out, which, usually, have widths less than ten microns, within the time of tens of picoseconds. This is our explanation of why LPI in HD is smaller than that in the traditional DD scheme due to thermally smoothing.

We have briefly added these clarifications in the revised manuscript.

11. Comment: *“Despite the mitigation relative to DD, the backscatter and hot-electron production in the HD scheme is still quite high and is likely to get worse when scaling up to a high-gain design. Are these levels believed to be acceptable for the HD design or are there plans for mitigation?”*

Response:

The fractions of the backscatter and the hot-electron production mainly depend on the parameters of the laser intensity I_L and the plasma environment of the density nonuniformities and density scale length L_n . In HD, the DD laser intensity of $I_L = 1.8 \text{ PW/cm}^2$, which serves for generating the supersonic-electronic-heat wave, and the ID corona plasma with radiation temperature of $T_r = 200 \text{ eV}$ are the same as in the future ignition target, but there is some difference in the density scale length $L_n \sim 300 \mu\text{m}$ for the experiment target and $\sim 400 \mu\text{m}$ for the designed ignition target. Therefore, we think the fractions of the backscatter and hot electrons in the present experimental target and the future ignition target should not differ much. In addition, the backscattering fraction of $\sim 5\%$, as shown in the recent NIF experiments, is acceptable for the ignition target physically and in laser energy.

For the hot electrons, we would like to explain whether they are able to preheat the fuel. The hot-electron range in the CH plasma can be written theoretically in the form $\rho R(\text{g/cm}^2) \approx 0.59 \times 10^{-5} E_e^{1.661}$ for $E_e > 2 \text{ keV}$ (Phys. Plasmas 18, 022703(2011)), where E_e is the hot-electron energy in units of keV. It is seen from the expression that even if the hot-electron energy is large enough, such as $E_e = 100 \text{ keV}$, the range is $\rho R \approx 0.012 \text{ g/cm}^2$. On the other hand, the simulations show that in the modeling ignition target, the area density of the remainder of the ablating CH ablator is about 0.06 g/cm^2 at the time of the maximal implosion velocity in which 0.02 g/cm^2 is from the density plateau piled up by the isothermal compression.

This indicates that the hot electrons are stopped in CH and prevented from preheating the fuel in the ignition target.

12. Comment: *What is the f-number of the drive beams, as this has a significant impact on SBS/SRs backscatter?*

Response:

The f-number on the SG-III laser facility is 10. From the results of the HD experiment, the influence of this f-number on LPI seems to be small.

REVIEWER COMMENTS

Reviewer #1 (Remarks to the Author):

I have read the revised version of the article and the answers to my comments from the authors. I am satisfied with their answers.

I could still have some minor doubts(e.g., the result obtained with the hemispherical taret is till a bit "marginal") but in general I believe that this contribution is important. In this sense I disagree with the comments of one of the other referees who consider this work "incremental". Indeed I think this is not incremental.

I have also looked ta the other comments raised by the other referees and the authors' replies, d again in general I find the answers correct, even if not always fully convincing.

In conclusion I would say that the manuscript can be published as it is

Reviewer #2 (Remarks to the Author):

This is my report on a review of a second submission of this article following revisions by the authors. In my previous report I raised concerns that the results reported are an incremental change to those already published in a high energy density physics (HEDP) paper. I suggested that the authors “re-write the manuscript to avoid repeating previously published work, and to identify the novel and high-impact results of the present work.”

In response, the authors provided a rebuttal that suggests that the new results provide experimental confirmation of the previous data (which I take to mean they have shown they are reproducible), and that provides a breakdown of three new insights that the new submission provides:

- Confirmed previous results at 3.6KJ indirect-drive laser energy and extended them with new experiments at 4.0KJ, providing more data to confirm pressure boosting with the HD scheme
- Demonstrated experimental changes which allow better scaling to ignition capable target designs
- Demonstrated the tuning of the DD laser to provide a “perfect” bulldozer effect and produce pressure boosting and smoothing

While the above statements are fairly clear, I don't see correspondingly clear statements in the updated manuscript, which to my mind significantly degrades the impact of the work.

The authors also responded to my comments by adding a series of new data using hemispherical ablator targets as a comparison with the planar targets in the previous draft. This is a welcome addition to the paper but does not directly address my concern

I still believe that the current submission lacks a proper discussion of the context of new data with respect to the previously published results. For example, the key result in the abstract of this paper is that “This article reports that... such a boosted and smoothed HD pressure is first verified experimentally” while the previous HEDP paper contains the statement “Thus, the experiments have verified the HD pressure boost compared to the ID pressure”. These statements suggest that the same conclusions are being presented in two separate works, which according to the rebuttal (not the manuscript) is not the case.

Given the more detailed description in this paper compared to the previous, and the potential high impact of the hybrid-drive scheme, I am prepared to recommend this paper for publication. However, I would like to stress that the addition of a short paragraph discussing how these experiments add to the understanding of the HD scheme will result in a higher quality article where the impact is more clear.

Reviewer #3 (Remarks to the Author):

Dear Editor,

For the most part I am satisfied with the authors’ response to the referees’ comments. I think the question of whether similar results being previously published as part of a conference proceedings precludes publication in Nature Comms is a decision for the Editor. There are a couple of points in the authors’ response that I am concerned about, but they are not critical to the main results:

1) The authors stated that electrons with energies less than 250 keV will be stopped in the CH ablator (and thus are not a preheat concern), but this is about an order of magnitude higher than the typical electron energies that are thought to be a preheat concern in direct-drive implosions.

2) I do not think that thermal smoothing in the HD scheme is the source of reduced hot-electron production relative to traditional DD. Even if there were significant density nonuniformities in the corona of a DD implosion, filamentation and self-focusing are effectively absolute instabilities, so an enhanced seed level would not have a significant impact on these instabilities (also the enhanced seed would have to be wavelength and phase matched to the beam-driven perturbation). That being said, it is not obvious to me why the hot-electron production is reduced in the HD scheme, but it could be as simple as the fact that the ID beams increase T_e , which would result in increased thresholds for

filamentation and TPD. Regardless of the reason, LPI is not of primary importance to these results, and this is not the appropriate forum to be introducing an LPI mitigation mechanism that (to my knowledge) has not been previously discussed in the literature without adequate supporting evidence.

Response Letter

Article ID: NCOMMS-22-52906

Title: Experimental confirmation of drive pressure boosting and smoothing for hybrid-drive inertial fusion at the 100-kJ laser facility

Authors: Ji Yan, Jiwei Li, X. T. He, et al.

We are grateful to the referees for their further insightful reviews and valuable comments on our revised manuscript. According to their comments, we have carefully revised the manuscript again. Following are the specific responses to the referees' comments point by point. With these responses, we believe that we have overcome all referees' criticisms and fulfilled the criteria set by them for accepting our manuscript for publication in Nature Communications. The changes in the revised manuscript are listed in the following.

I. Responses to Referee #1

Comment: *"In conclusion I would say that the manuscript can be published as it is"*

Response: Thanks for the referee's recommendation of our manuscript for publication.

II. Responses to Referee #2

Comment 1: *the key result in the abstract of this paper is that "This article reports that... such a boosted and smoothed HD pressure is first verified experimentally" while the previous HEDP paper contains the statement "Thus, the experiments have verified the HD pressure boost compared to the ID pressure". These statements suggest that the same conclusions are being presented in two separate works, which according to the rebuttal (not the manuscript) is not the case.*

Response: The previous result using a planar ablator target at DD laser energy of 3.6 kJ to experimentally confirm HD pressure boosting for the first time is obviously important, but indeed applying the term "verify" is not strict enough because there is only one experimental result without more repeated or similar results to support it. In the present manuscript, new more sufficient data from the hemispherical and planar ablator targets at DD laser energy of 3.6-4 kJ are provided to fully confirm HD pressure boosting and smoothing, while among these data, the experiment on the planar target at DD laser energy of 3.6 kJ is for "check and further confirm the previous result", which has been mentioned all in the abstract, the planar target experiment at DD laser of 3.6 kJ, and the section of "Discussions and conclusions".

Comment 2: *I would like to stress that the addition of a short paragraph discussing how these experiments add to the understanding of the HD scheme will result in a higher quality article*

where the impact is more clear.

Response: In the present manuscript, we have already added explanations of how these experiments increase understanding of the HD scheme in the section "Discussions and Conclusions"

III. Responses to Referee #3

Comment 1: *The authors stated that electrons with energies less than 250 keV will be stopped in the CH ablator (and thus are not a preheat concern), but this is about an order of magnitude higher than the typical electron energies that are thought to be a preheat concern in direct-drive implosions.*

Response: In the last response to comment 11 of reviewer #3, we used hot electron energy of 100 keV, from which with the expression of $\rho R(\text{g/cm}^2)$ we estimated the area density (the penetrating range) to be 0.012 g/cm^2 less than 0.06 g/cm^2 for the remaining CH ablator at the time of maximal implosion velocity. As for energy 250 keV mentioned in the response to the comment to reviewer #2, we mean that for hot electrons with energy of 250 keV the area density is just equal to the remaining CH area density of 0.06 g/cm^2 , and therefore, only these electrons with energies greater than 250 keV can penetrate the remaining CH preheating the fuel or to say that those electrons with energies less than 250 keV are trapped in the remaining CH.

In the ID scheme, the experiment on hot electron preheating fuel in the ignition-scale target for NIC (PRL 108, 135006(2012)) has shown that with ID laser of 1.3 MJ, the energy of energetic electron $> 100 \text{ keV}$ deposited into the CH ablator is about $570 \pm 250 \text{ J}$, and $5 \pm 3 \text{ J}$ (an upper bound) is absorbed in the DT ice, showing an acceptable increase of 3.5% for the adiabat of $\alpha = 1.5$. For the HD scheme, energetic electron preheating fuel needs further to be investigated.

Comment 2: *I do not think that thermal smoothing in the HD scheme is the source of reduced hot-electron production relative to traditional DD. Even if there were significant density nonuniformities in the corona of a DD implosion, filamentation and self-focusing are effectively absolute instabilities, so an enhanced seed level would not have a significant impact on these instabilities (also the enhanced seed would have to be wavelength and phase matched to the beam-driven perturbation). That being said, it is not obvious to me why the hot-electron production is reduced in the HD scheme, but it could be as simple as the fact that the ID beams increase T_e , which would result in increased thresholds for filamentation and TPD. Regardless of the reason, LPI is not of primary importance to these results, and this is not the appropriate forum to be introducing an LPI mitigation mechanism that (to my knowledge) has not been previously discussed in the literature without adequate supporting evidence.*

Response: In the last response to the reviewer's comment, we mentioned that in the DD

scheme, the DD laser with the intensity of 1.8 PW/cm^2 provides a nonuniform electron corona plasma, which results in filamentation and self-focusing increase to make the DD laser intensity over 1.8 PW/cm^2 forming new laser speckles in filamentation and self-focusing, resulting in a new LPI source, here **LPI refers to the laser plasma instability of SRS, SBS, TPD, etc., not only the filamentation and self-focusing itself**. While for the HD scheme, the ID laser provides a more uniform ID corona plasma with local equilibrium temperature $T_r = T_e = T_i = 200\text{eV}$, which results in filamentation and self-focusing mitigation, and therefore, LPI reduction by thermal smoothing compared with the DD scheme, here LPI, of course, involves SRS, SBS, and TPD, etc. This phenomenon is indeed seen in the comparative study by numerical simulation in which SBS and SRS occur rapidly with the formation of filamentations when the DD laser with the intensity of 1.8 PW/cm^2 is incident on a nonuniform plasma, but these are not clear in the uniform plasma.

We think in a novel HD scheme different from DD and ID schemes, how the thermodynamic equilibrium ID plasma affects LPI is still understudied not too much. In our report, we mainly investigate pressure smoothing and smoothing, which is key physics in the HD scheme, and there is no need for further discussion of the LPI mitigation mechanism. We thank the reviewer's comment: "LPI is not of primary importance to these results, and this is not the appropriate forum to be introducing an LPI mitigation mechanism that has not been previously discussed in the literature without adequate supporting evidence". At the end of the section "Fractions of stimulated backscattering and hot electron energy" in the present manuscript, we have revised the previous discussions related to the LPI mitigation mechanism to change into the statement "So far, why the LPI in the HD scheme is smaller than the LPI in the DD scheme that has no radiation ablation is still not very clear, which needs further exploration and discussion."

IV. List of changes

IV. List of changes

1. In response to comment 1 from reviewer #2, in addition to explaining the previous result in this Response Letter, in the present manuscript, we also stated it in red font in the **Abstract**, the section "**Radiation temperature and ID and DD velocities**", and the section "**Discussions and Conclusions**".

In response to comment 2 from reviewer #2, we added explanations of how these experiments increase understanding of the HD scheme in the section "Discussions and Conclusions" in red font.

2. In response to comment 2 from reviewer #3, in the present manuscript at the end of the section "**Fraction of stimulated backscattering and hot-electron energy**", we revised the previous discussions related to the LPI mitigation mechanism to change into the statement "So far, why the LPI in the HD scheme is smaller than the LPI in the DD scheme that has no radiation ablation is still not very clear, which needs further exploration and discussion" in red font.

REVIEWERS' COMMENTS

Reviewer #2 (Remarks to the Author):

The present manuscript has addressed my detailed questions, and I believe the changes have put the results of this work in context with previous publications. I believe that the current form of the manuscript is suitable for publication.

Reviewer #3 (Remarks to the Author):

I am satisfied with the Authors' response and recommend the manuscript for publication.

REVIEWERS' COMMENTS

Reviewer #2 (Remarks to the Author):

The present manuscript has addressed my detailed questions, and I believe the changes have put the results of this work in context with previous publications.

I believe that the current form of the manuscript is suitable for publication.

Reviewer #3 (Remarks to the Author):

I am satisfied with the Authors' response and recommend the manuscript for publication.

Response to reviewer' s comment

We are grateful for the reviewers' acknowledgement of our work. Thanks all the reviews for their contribution to improving the quality of the paper and promotion of our understanding of the work.